# Individual variation in Achilles tendon morphology and geometry changes susceptibility to injury

**Nai-Hao Yin[1], Paul Fromme[2], Ian McCarthy[3], Helen L Birch[1]\***

[1]Research Department of Orthopaedics and Musculoskeletal Science, University College London, Royal National Orthopaedic Hospital, Stanmore, United Kingdom; [2]Department of Mechanical Engineering, University College London, London , United Kingdom; [3]Pedestrian Accessibility and Movement Environment Laboratory, Department of Civil, Environmental and Geomatic Engineering, University College London, London, United Kingdom

**Abstract** The unique structure of the Achilles tendon, combining three smaller sub-tendons, enhances movement efficiency by allowing individual control from connected muscles. This requires compliant interfaces between sub-tendons, but compliance decreases with age and may account for increased injury frequency. Current understanding of sub-tendon sliding and its role in the whole Achilles tendon function is limited. Here we show changing the degree of sliding greatly affects the tendon mechanical behaviour. Our in vitro testing discovered distinct sub-tendon mechanical properties in keeping with their mechanical demands. In silico study based on measured properties, subject-specific tendon geometry, and modified sliding capacity demonstrated age-related displacement reduction similar to our in vivo ultrasonography measurements. Peak stress magnitude and distribution within the whole Achilles tendon are affected by individual tendon geometries, the sliding capacity between sub-tendons, and different muscle loading conditions. These results suggest clinical possibilities to identify patients at risk and design personalised rehabilitation protocols.

**\*For correspondence:**
h.birch@ucl.ac.uk

**Competing interests:** The authors declare that no competing interests exist.

## Introduction

The Achilles tendon is fundamental to human locomotion, and its mechanical properties are an important contributor to athletic ability. Although the Achilles tendon is the strongest tendon in the human body and solely responsible for powering ankle extension in 'push off' (*Finni et al., 1998*), the Achilles tendon is often the Achilles heel in athletes and also in the general adult population. The Achilles tendon has a spring-like function (*Lichtwark and Wilson, 2007*), and the timely release of elastic energy is important to facilitate forward body propulsion. The tendon spring operates optimally at relatively high strains as this increases the utilisation of elasticity (*Lichtwark, 2005*; *Lichtwark and Wilson, 2006*), but as a consequence, the risk of developing tendon injuries, such as chronic painful tendinopathy or acute tendon ruptures, increases (*Magnan et al., 2014*; *Martin et al., 2018*). Allowing for a high degree of elongation during numerous cycles of loading and unloading is challenging for a biological material, particularly given the complex twisted structure of the Achilles tendon.

The human Achilles tendon, unlike the majority of tendons, can be separated into three distinct sub-tendons (*Handsfield et al., 2016*) instead of one uniform common tendon (*Edama et al., 2015*; *Szaro et al., 2009*). Sub-tendons arise from the lateral and medial heads of the gastrocnemius and the soleus muscles. The fascicles (the primary load bearing structure of tendon) within each sub-tendon do not intertwine until reaching the insertion site on the calcaneus. Achilles tendon morphology

is highly individualised with different geometries at both whole- and sub-tendon levels, including differences in cross-sectional areas (CSA) and twist angles of sub-tendons (*Edama et al., 2015*; *Pękala et al., 2017*). These anatomical features support the idea that each sub-tendon should be regarded as an individual functional unit within the whole Achilles tendon in order to fully understand the internal force distribution. Current medical imaging modalities however have yet to develop sufficient resolution to accurately visualise the borders of sub-tendons in vivo. As a result, there is limited understanding of the relationship between sub-tendon morphology, mechanical behaviour, and injury risk, and a lack of translation from anatomical knowledge to clinical treatment options.

Although retaining the separation of sub-tendons along the length of the Achilles tendon poses mechanical challenges, having individual control of three sub-tendons is likely to be beneficial. At the distal end of the muscle-tendon-bone unit, the joints in the human foot and ankle are highly mobile to ensure firm contact with the ground in uneven terrains. Proximally, gastrocnemii and soleus muscles show distinct muscle fibre characteristics (*Gollnick et al., 1974*) and activate differently to serve their distinct functions in human movement (*Cronin et al., 2013*). These features corroborate the idea that having individual control of sub-tendons improves movement efficiency. For this optimised control to occur, the interface among sub-tendons must allow a certain degree of sliding (*Handsfield et al., 2017*; *Sun et al., 2015*), creating different internal displacement gradients. Optimal sliding of loaded sub-tendons may help distribute stress to other unloaded sub-tendons reducing the risk of overload injury (*Maas and Finni, 2018*), but sliding may also create localised shear stress and strain at the interfaces, which is likely to be detrimental (*Kondratko-Mittnacht et al., 2015*; *Scott et al., 2015*). The physiological advantages and the consequences of excessive or reduced sub-tendon sliding are not understood, and simple, straightforward interpretation should be avoided due to the complexity of sub-tendon geometry, which differs in CSA and twist angle.

Non-uniform displacement within the Achilles tendon has been widely demonstrated under various static (*Arndt et al., 2012*; *Bogaerts et al., 2017*; *Stenroth et al., 2019*) and dynamic (*Chernak Slane and Thelen, 2014*; *Franz et al., 2015*; *Franz and Thelen, 2015*; *Slane and Thelen, 2015*) loading conditions. This non-uniformity decreases as people age (*Clark and Franz, 2020*; *Franz and Thelen, 2015*; *Slane and Thelen, 2015*) and after repair of a ruptured tendon (*Beyer et al., 2018*; *Fröberg et al., 2017*). There is also a trend towards more uniform displacement in tendons showing signs of pathological changes (*Couppé et al., 2020*). Reduced sub-tendon sliding therefore seems to negatively correlate with tendon function. Further evidence for the advantage of sliding within the tendon substructure of energy-storing tendons comes from the equine superficial digital flexor tendons (SDFT). Age-related reduction in sliding ability has been reported at the fascicle level in the equine SDFT (*Thorpe et al., 2013*; *Thorpe et al., 2012*) and is believed to negatively affect energy-storing capacity and increase risk of injuries. Most in vivo human studies of strain distribution have used ultrasound speckle tracking techniques to observe intra-tendinous movement. The main limitation of this technique is the limited plane of view and the lack of visualisation of twisted and highly variable internal sub-tendon boundaries. Although it is convenient to assume that the superficial part of the Achilles tendon is composed largely of fascicles arising from gastrocnemii and the deeper part from fascicles from the soleus, this likely to be a considerable source of error as the actual orientation differs between individuals. This limitation can be overcome by employing finite element analysis (FEA) on subject-specific tendon geometry (*Hansen et al., 2017*) to systematically study the complex interaction between interface sliding property (*Handsfield et al., 2017*) and the force output from the associated muscles bellies.

In this study, we have conducted a series of multi-modality experiments to understand sub-tendon behaviour within the Achilles tendon. First, we dissected five human Achilles tendons into their sub-tendon components and performed in vitro mechanical tests, respectively, in order to compare their material properties. Second, a further three Achilles tendons were dissected, and precise morphology and geometry were recorded. In silico models were generated using the measured geometries and material properties. We used FEA to explore how different sliding properties affect sub-tendon displacement and stress distribution and the variation between different models. Particular interest was paid to the proximal soleus sub-tendon face since this location is an anatomical landmark, which can be clearly observed and quantitatively measured using traditional ultrasonography. Finally, we conducted an in vivo study combining non-invasive muscular stimulation and

ultrasonography to compare the modelling results with experimental data, particularly with respect to the impact of age-related decrease in non-uniform displacement within the Achilles tendon. These results provide a new paradigm for understanding human Achilles tendon function and its injury mechanism.

## Results

### Sub-tendons within the Achilles tendon have different mechanical properties

Five (two males and three females, aged 78–87 years) fresh-frozen human Achilles tendon specimens were carefully dissected, and individual sub-tendons were subjected to quasi-static mechanical tests to failure. The soleus sub-tendon had the greatest CSA and the highest failure force and stiffness compared to both gastrocnemii sub-tendons (*Table 1*, detailed statistical analysis results in *Supplementary file 1*). There were no significant differences in material properties between the sub-tendons.

### Sub-tendons have different CSA, shapes, and twists

Another three Achilles tendons (age–sex: 54-M, 55-M, and 14-F – constructed into Models 1, 2, and 3, respectively) were collected and dissected into sub-tendon components. These tendons were selected based on availability and to represent a diverse range of individual differences in tendon morphology. The precise morphology and geometry were recorded and used to create 3D computer-aided design models. The tendons varied in CSA from 42.3 to 90.4 $mm^2$, and the twist and the internal arrangement of their sub-tendons showed great individual variability (*Figure 1*), as reported previously (*Edama et al., 2015*; *Pękala et al., 2017*). In the proximal part of the tendon, the anterior portion was mostly occupied by the soleus sub-tendon and the posterior portion by medial gastrocnemius on the medial aspect and lateral gastrocnemius on the lateral aspect. Passing distally, the sub-tendons rotate in a lateral direction and the longest tendon (Model 1, 70 mm) rotated the most since its lateral gastrocnemius sub-tendon almost completely occupied the anterior surface. In the shorter tendons (Models 2 and 3, 40 mm), at the distal end, the anterior portion was occupied by the lateral gastrocnemius and soleus sub-tendons. The tendon used for Model 3 was the thinnest (CSA: 42.3 $mm^2$), while tendons used in Models 1 and 2 were substantially thicker (89.1 and 90.4 $mm^2$). The models closely replicated the original sub-tendon geometries, capturing individual variability in sub-tendon arrangements. The tendons (and models) are not intended to represent the geometrical features of a particular gender or age group, but to represent diversity among individuals.

**Table 1.** Mechanical testing results of three Achilles sub-tendons.

| Specimen | CSA ($mm^2$) | | | Failure force (N) | | | Ultimate stress (MPa) | | | Ultimate strain (%) | | | Young's modulus (MPa) | | | Stiffness (N/mm) | | |
|---|---|---|---|---|---|---|---|---|---|---|---|---|---|---|---|---|---|---|
| (age–sex) | LG | MG | S | LG | MG | S | LG | MG | S | LG | MG | S | LG | MG | S | LG | MG | S |
| 69 M | 13.6 | 14.5 | 39.2 | 325.0 | 402.2 | 1701.2 | 24.0 | 27.7 | 45.3 | 9.7 | 10.0 | 19.1 | 353.9 | 375.8 | 399.4 | 79.9 | 97.6 | 260.6 |
| 78 F | 12.4 | 10.8 | 20.6 | 154.7 | 571.1 | 1155.6 | 12.5 | 52.6 | 56.2 | 11.6 | 8.9 | 17.5 | 175.0 | 771.1 | 421.1 | 36.7 | 145.6 | 167.8 |
| 84 F | 5.7 | 10.7 | 30.9 | 503.6 | 621.8 | 1577.0 | 89.0 | 58.3 | 51.0 | 8.9 | 12.2 | 10.4 | 1231.5 | 636.9 | 734.9 | 118.2 | 121.4 | 385.1 |
| 85 F | 10.4 | 14.0 | 38.2 | 787.5 | 688.3 | 1090.9 | 75.8 | 49.0 | 28.6 | 11.9 | 10.6 | 15.8 | 853.6 | 617.2 | 239.9 | 151.7 | 144.7 | 156.6 |
| 87 M | 4.6 | 17.0 | 35.2 | 301.6 | 647.7 | 1578.5 | 65.9 | 38.1 | 44.9 | 8.2 | 10.2 | 10.8 | 1033.4 | 496.2 | 606.2 | 80.9 | 143.0 | 349.6 |
| Mean | 9.3* | 13.4* | 32.8 | 414.5* | 586.2* | 1420.6 | 53.4 | 45.1 | 45.2 | 10.1 | 10.4 | 14.7 | 729.5 | 579.4 | 480.3 | 93.5* | 130.5* | 263.9 |
| SD | 4.0 | 2.7 | 7.6 | 242.5 | 111.3 | 277.1 | 33.4 | 12.2 | 10.4 | 1.6 | 1.2 | 3.9 | 449.5 | 149.9 | 192.7 | 43.5 | 20.9 | 103.4 |

CSA: cross-sectional area, LG: lateral gastrocnemius, MG: medial gastrocnemius, S: soleus sub-tendon. *Significantly different from soleus sub-tendon after post hoc analysis (p<0.017).

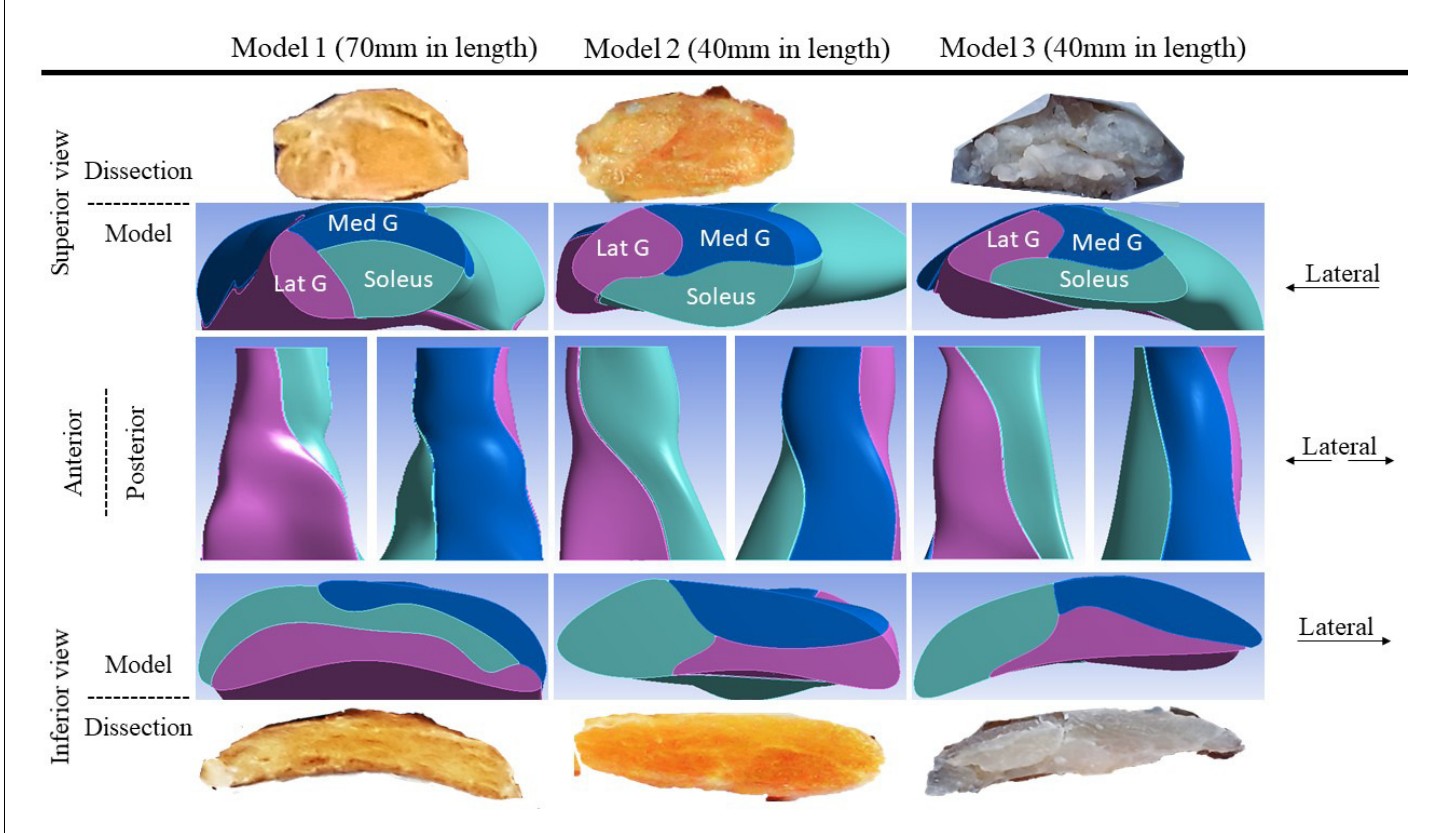

**Figure 1.** Three-dimensional computer models of Achilles sub-tendons.

## Sub-tendon displacement is affected by the degree of sliding but is geometry dependent

To study the effect of sub-tendon loading on the longitudinal displacement of the proximal soleus face, we ran an FEA of each model with the same axial tension (100 N) in each sub-tendon but with different degrees of sliding between them – namely, frictionless, frictional (with friction coefficient 0.2, 0.4, 0.6, 0.8, and 1.0) and bonded (no sliding). When the soleus sub-tendon was loaded in isolation, assigning frictional contacts reduced the soleus displacement compared to the frictionless contact across all three models. When the medial gastrocnemius sub-tendon was loaded, soleus displacement increased with increasing friction. This trend was also observed when the lateral gastrocnemius was loaded, but interestingly only in Models 1 and 2 (*Figure 2*). When loaded, the three models showed distinct transverse plane rotation behaviour, which is likely caused by their different geometrical features (*Figure 2—figure supplement 1*).

Displacement data were normalised to frictionless contact conditions to observe the relative displacement change and therefore the impact of increasing cohesion between the sub-tendons, as may occur with ageing (*Figure 3*). When the soleus was loaded, all three models showed a similar decrease in soleus displacement as friction increases. In the gastrocnemii-loaded conditions, a substantial increase in soleus displacement was observed in Model 1 and a lesser increase was observed in Model 2, suggesting tendon shapes and geometries affect force transmission across sub-tendons when interface properties change.

## Reduced sliding decreases mean stress of the whole tendon, but not peak stress

All models showed decreased mean von Mises stress for frictional and bonded contacts compared to frictionless contact (*Figure 4*, upper row); however, the peak von Mises stress, a prediction of material yielding, differed among all three models (*Figure 4*, lower row). As friction increased, the

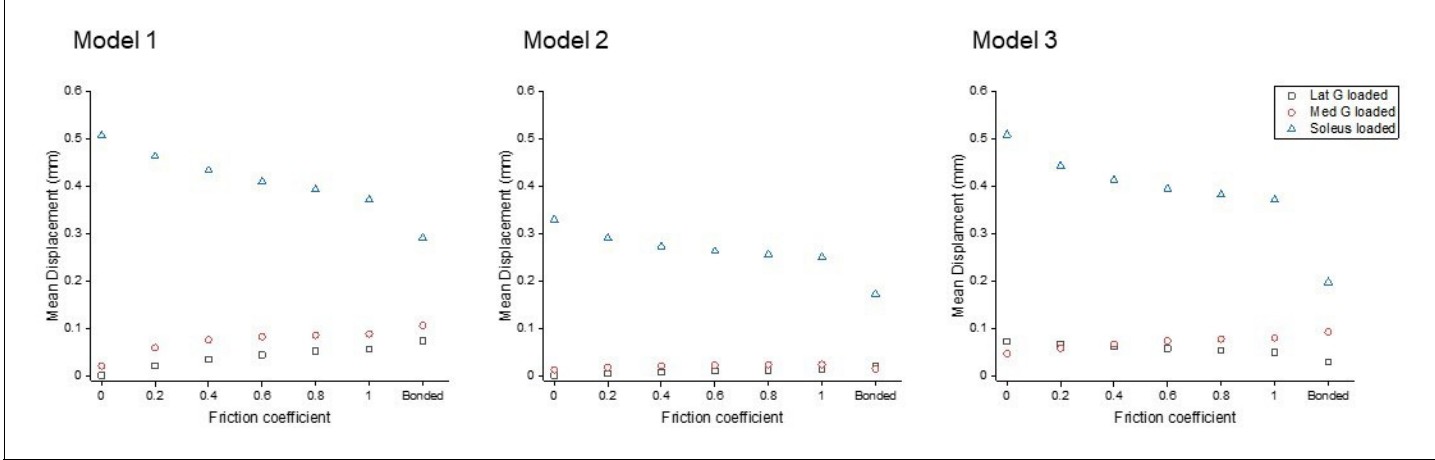

**Figure 2.** Mean displacement of the proximal soleus face of three models with different friction contacts when each sub-tendon was loaded in isolation. The online version of this article includes the following figure supplement(s) for figure 2:

**Figure supplement 1.** Mean transverse plane displacements (in mm) of the proximal soleus face of the three models with different friction contacts (x-axes) when each sub-tendon was loaded in isolation.

peak stress intensity of Model 1 decreased, but Models 2 and 3 fluctuated with different sub-tendon loading conditions. For bonded contact conditions, all models showed lower peak stress compared to the frictionless and frictional contact conditions, but the size of the decrease varied greatly between models. Furthermore, changing the contact properties, in addition to affecting the intensity of peak stress, also changed the region where the peak stress occurred (*Figure 5*). Due to the large geometry differences (CSA and twist) across the three models, no clear pattern in the change of peak stress location is seen.

## In vivo age-related displacement decline can be explained by modelling results

We further explored whether our modelling results (*Figure 3*) reflect in vivo age-related decline in displacement non-uniformity. Each muscle was electrically stimulated, and the resulting soleus musculotendinous junction displacements were recorded and normalised to their relative muscular displacements. The older group of participants, aged 52–67 years (n = 7), demonstrated significantly

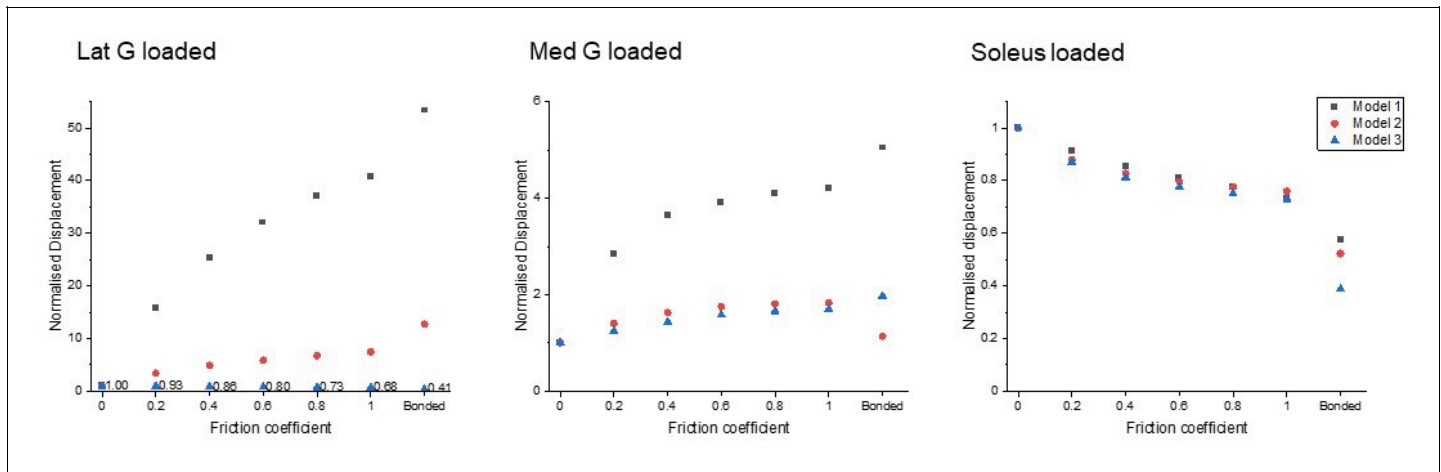

**Figure 3.** Normalised (to frictionless, coefficient = 0) proximal soleus face displacement of different friction contacts when each sub-tendon was loaded in isolation. Note the scale of the y-axes differs to allow better visualisation of differences in the data.

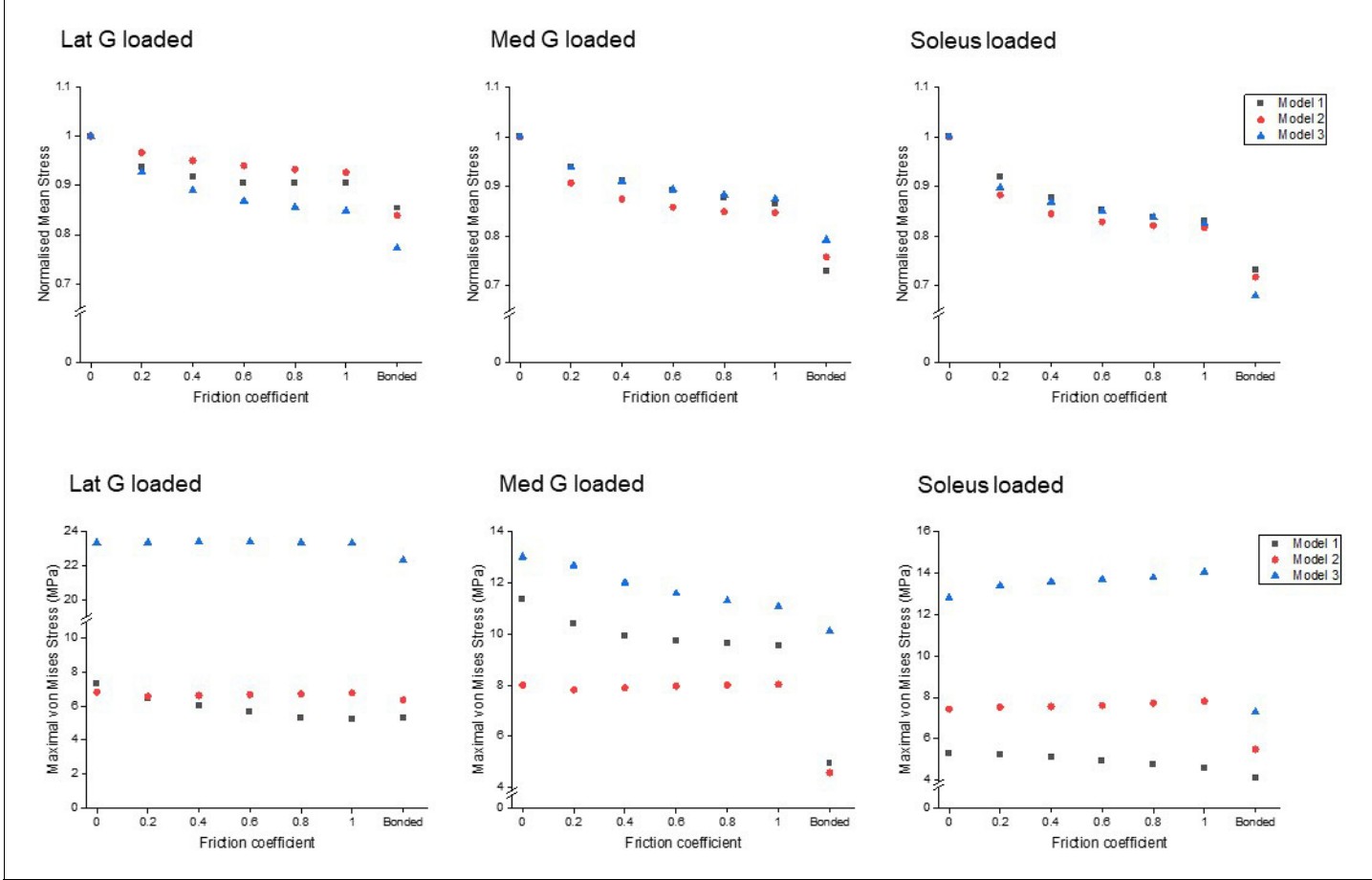

**Figure 4.** Mean normalised (upper row) and peak (lower row) von Mises stress of the whole Achilles tendon for different friction coefficients when each sub-tendon was loaded in isolation. Note the scale of the y-axes differs to allow better visualisation of differences in the data.

(p=0.005) lower normalised soleus junction displacement compared to the younger group (20–29 years, n = 9) for soleus stimulation (*Figure 6*). This decrease in displacement with increasing age shows the same trend as the modelling result when the soleus sub-tendon was loaded in isolation (*Figure 3*, right). We found no detectable differences in soleus junction displacement between groups for both gastrocnemii stimulation conditions (p=0.963 and p=0.558).

## Discussion

Our results demonstrate that the mechanical behaviour of the Achilles tendon is highly complex and affected by a combination of factors including different sub-tendon mechanical properties, individual tendon morphology, and age-related changes in sub-tendon sliding. We suggest that radical changes are needed to the way in which Achilles tendon mechanics are studied, and also in the way in which we think about factors leading to the development of tendon injuries and rehabilitation.

Different mechanical properties between sub-tendons are due to differences in CSA and most likely relate to the different properties of their connected muscle bellies. As a result of high mass and relatively short muscle fibre length, the soleus muscle has the greatest force-generating capacity and can produce over half of the total ankle plantar flexion torque (*Dayton, 2017*). Reflecting this large difference in force-generating capacity between gastrocnemii and soleus, the soleus sub-tendon had the greatest CSA and highest failure force and stiffness among the three sub-tendons. Stiffness could therefore scale with muscular force and CSA among different sub-tendons. It is generally agreed that gender differences exist when comparing tendon mechanical properties, and although not designed to test this, our study showed similar trend with male specimens having higher

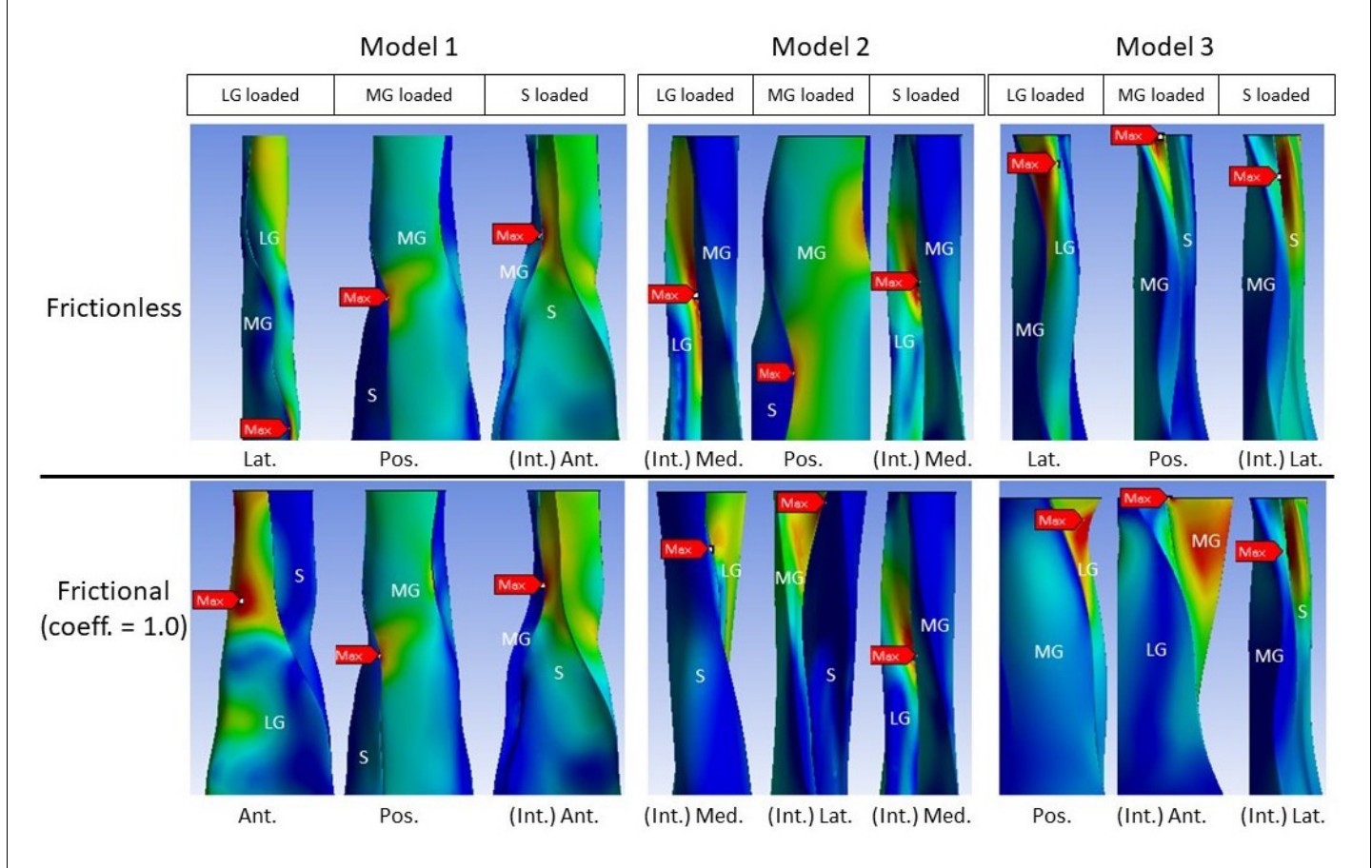

**Figure 5.** Change in peak stress location when shifting from frictionless (upper row) to frictional (lower row) contact is structure dependent. LG: lateral gastrocnemius, MG: medial gastrocnemius, S: soleus sub-tendon. View planes: Ant.: anterior, Pos.: posterior, Lat.: lateral, Med.: medial, Int.: internal view with the covering sub-tendon removed.

averaged combined CSA (M: 62.1 mm$^2$, F: 51.2 mm$^2$) and tendon failure force (M: 2478.1 N, F: 2383.5 N). Future studies including a larger sample size with detailed exercise history could provide further insight into gender differences in sub-tendon mechanics. Our measured Young's modulus of sub-tendons was approximately two-thirds of that previously reported for the whole Achilles tendon (*Wren et al., 2001*). This lower value for material stiffness of the sub-units compared to the whole structure may be an artefact of dissection and mechanical testing in isolation, as has been seen previously when testing tendon fascicles isolated from whole tendons (*Thorpe et al., 2012*), or due to differences in specimen fixation methods and loading protocols. The inconsistency may also be a result of differences in the age groups studied, as the study of whole Achilles tendons (*Wren et al., 2001*) found a negative correlation between material properties and age and had significantly more younger specimens comparing to our relatively old samples. It is generally agreed, but inconclusive, that the in vivo modulus and strength of Achilles tendon decreases with age (*Svensson et al., 2016*); however, studies have yet to measure the interface property and sub-tendon morphology when studying individual mechanical properties, ignoring two important confounding factors as our modelling result demonstrated.

Our modelling results provide insight into the impact of different sub-tendon shapes and orientations on sub-tendon sliding. Furthermore, our results highlight the importance of understanding sub-tendon sliding when considering the non-uniform displacement of Achilles tendon during movements (*Handsfield et al., 2017*). We were able to study this by assigning frictional contacts and varying the friction coefficient in our FEA to represent more closely the physiological sub-tendon behaviour and changes across the lifespan, instead of the dichotomy between frictionless sliding and

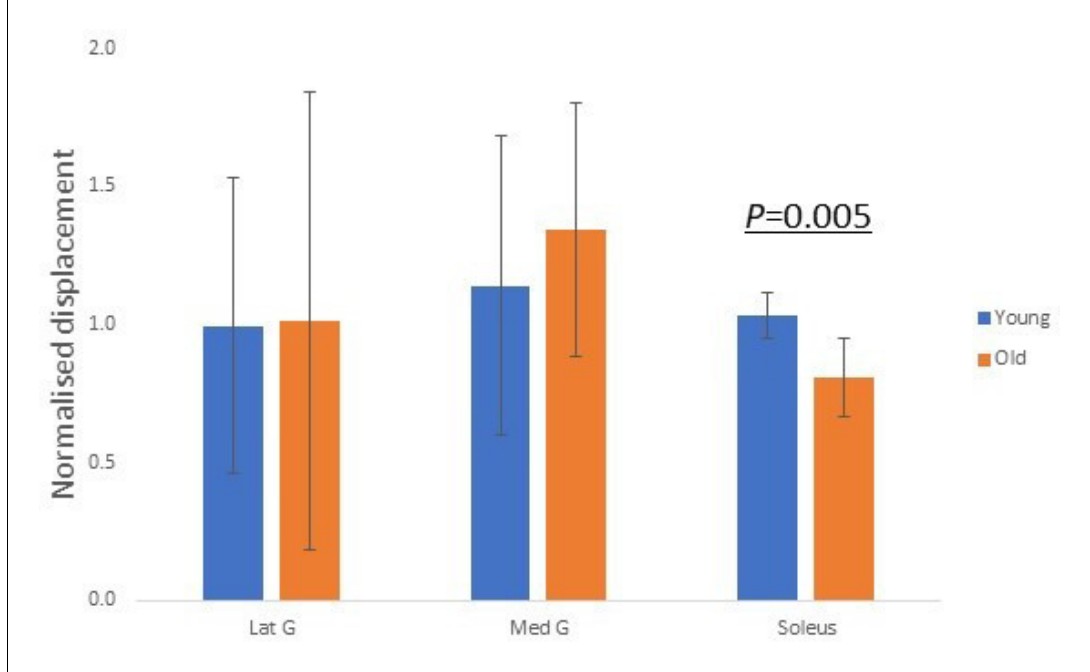

**Figure 6.** Group differences in normalised soleus junction displacement during different stimulation trials.

completely bound conditions as has been applied previously (*Handsfield et al., 2017*). We arbitrarily assigned identical coefficient across three interfaces, but the true sliding ability between sub-tendons and the rate of age-related decline could be different. An increase in friction in our model, mimicking the age-related reduction in sliding under physiological loads (*Thorpe et al., 2013*), greatly affected the sub-tendon displacement and stress distribution. This was highly dependent on geometry and varied between the three models, suggesting the rotation and twist angle of the whole Achilles and sub-tendons affect how they interact (*Figure 2—figure supplement 1*). When increasing the friction between sub-tendons, we would expect to see an increase in the influence of individual muscle belly contractions on adjacent sub-tendons. In keeping with this, the degree of soleus displacement due to gastrocnemius applied load mainly increased, but this was much more evident in the longest model with the greatest sub-tendon twist (Model 1) and least in the shorter model with the smallest CSA (Model 3). A greater transfer of load between the gastrocnemius and soleus would be expected to result in reduced soleus displacement when the soleus muscle alone applies force. All three morphologies studied showed reduced soleus displacement with simulated soleus contraction as the coefficient of friction was increased. However, contrary to the above, our studies showed the least reduction in Model 1 and the greatest reduction in Model 3. These results highlight the complexity of the tendon and suggest that sub-tendon shape, twist, and rotation results in unequal distribution of strain along the length of the tendon.

Similarly, when friction increased, the effect on peak stress intensity differed between the models. In the most twisted model (Model 1), an increase in friction reduced the peak stress intensity, while in the other models, peak stress intensity fluctuated under different muscle loading conditions. These results strongly suggest that the impact of age-related changes to the stiffness of the interface depends on individual Achilles tendon morphology and balance of muscle strength. Load sharing, such as between adjacent muscle-tendon units (*Maas and Finni, 2018*) and between fascicles (*Thorpe et al., 2013*), has been proposed as a strategy to minimise stress and lower the risk of injury. As we demonstrate here however, the reduction of the whole tendon stress does not necessarily translate to a reduction in the peak stress (*Figure 4*), which is more important when considering the strength and integrity of a material under load. The high-stress region during loading could represent a mechanical weak point that may later develop pathological changes. It has been demonstrated that stress distribution within the Achilles tendon is geometry dependent and, interestingly, changing the material properties had minimal effect on tendon stress distribution (*Hansen et al.,*

*2017*). In addition, our results further imply that certain types of tendon shape or twist have a higher injury risk factor and may predispose to age-related injury. Our study is limited to three different morphologies, but it is likely that the range of morphologies is far greater than that modelled here, and this would result in an even more disparate strain and stress distributions between individuals.

The results of our study suggest that using the traditional anatomical landmark (soleus musculotendinous junction, the start of the free Achilles tendon) to measure in vivo tendon mechanical properties (*Kongsgaard et al., 2011*) is problematic, especially when studying aged or repaired Achilles tendons. The displacement of the soleus musculotendinous junction is used in in vivo studies to calculate tendon strain and stiffness; however, our study has shown that displacement is not equivalent or proportional to whole tendon strain when comparing tendons with different morphologies and sub-tendon arrangements. In low-friction conditions, simulating a young person's tendon, the displacement of the soleus junction is predominately from the soleus muscle-(sub-)tendon unit displacement; in high-friction conditions, mimicking aged tendons, the displacement of the soleus junction is the combination of both the decreasing trend from soleus and the increasing influence from gastrocnemii muscle-tendon units. Moreover, the exact geometry and mechanical properties of each sub-tendon among individuals are unknown, and each associated muscle exhibits different age- and injury-related compositional (*Csapo et al., 2014*) and structural (*Stenroth et al., 2012*) changes. Together, we suggest that measuring 'whole' tendon properties is insufficient when investigating such tendons. Previous studies have shown that repaired Achilles tendons show higher displacements when loaded compared to healthy tendons, despite being thicker and transmitting lower forces (*Geremia et al., 2015*; *Wang et al., 2013*). Considering that those sub-tendons are surgically bound together and show uniformed displacements (*Beyer et al., 2018*; *Fröberg et al., 2017*), the actual decline in material properties post-surgery may be even greater than reported since sub-tendons exhibit much lower displacement when bound, given the same force and material properties.

In our in vivo studies, we found a negative relationship between participant age and the electrical-induced soleus junction displacements, and this relationship supported our FEA results when assigning frictional contacts, reinforcing the belief that the matrix between sub-tendons stiffens with age. No statistically significant difference was detected between young and old groups in both gastrocnemii stimulation trials. Our modelling results suggested that gastrocnemii-induced soleus displacement is generally low and, importantly, depends on the shape and twist of the tendon, which we were unable to measure in our in vivo studies. Furthermore, the gastrocnemii have longer tendinous parts that have complex interactions with the underlying soleus muscle and aponeurosis (*Finni et al., 2003*; *Magnusson et al., 2003*), and the CSA of gastrocnemii sub-tendons are smaller and may have less influence on the larger soleus sub-tendon. The simplification of our in vivo study may therefore overlook certain age-related features in muscle-tendon units and surrounding structures that may potentially affect our results. To the best of our knowledge, no study has investigated whether material properties of sub-tendons reduce with age or whether the reduction is similar among three sub-tendons; thus, we cannot conclude that the measured age-related decline in displacement is solely from the reduction of interface sliding ability without the influence from altered tendon material properties. Our use of minimal electrical intensity during stimulation prevented co-contraction of other muscles, but the expected contraction force was low and less than most physiological loading scenarios. Overall, our in vivo method provides a relatively simple, low-cost setup that is able to detect in vivo sub-tendon sliding ability. In future studies combining this method with other techniques such as ultrasound speckle tracking, shear-wave elastography (*Lima et al., 2018*), skin advance-glycation end-product measurements (*Couppé et al., 2014*), or Raman spectroscopy (*Yin et al., 2020*) may enable non-invasive measurement of 'tendon age', which is a valuable but currently unobtainable clinical parameter in studying tendon injuries.

## Conclusions

In conclusion, our study provides novel concepts and improved understanding of Achilles tendon mechanical behaviour. The different mechanical properties of Achilles sub-tendons could originate from various structural or compositional adaptations (*Birch, 2007*) to different functional demands. Our results suggest that using the musculotendinous junction displacement to measure Achilles tendon strain in vivo should be done with caution and should not be used to compare tendons from different individuals without regard for tendon morphological variations such as CSA, shape, and twist. Differences in the transfer of force between sub-tendons with different geometrical arrangements

suggest that some tendon morphology types have a higher risk of injury and that sites within the tendon most prone to injury will vary between individuals. Our study has provided new insights into tendon mechanics and could inspire new, potentially personalised, treatment and prevention strategies to revolutionise current management of Achilles tendon health, injury prevention, and rehabilitation.

## Materials and methods

### Separation of Achilles sub-tendons

The tendon specimens (total n = 8) used in this study were collected from the Vesalius Clinical Training Centre, University of Bristol (REC 08/H0724/34) and from UCL/UCLH Biobank for Studying Health and Disease (HTA license number 12055) with Local R and D approval (Ref: 11/0464). Fresh-frozen specimens (from the Vesalius Clinical Training Centre, n = 5) were used for mechanical testing, and fresh specimens (from UCL/UCLH Biobank, n = 3) were used for morphology measurements and computer model construction. Only tendons showing no signs of injury or disease were included in the study. Sub-tendons were separated as described in the study (*Szaro et al., 2009*). For the three specimens used for model construction, a longitudinal reference line (from the mid-point of calcaneal insertion site to mid-point between the two gastrocnemii bellies) was drawn before separation as a guide to align tendon cross-sections (see below). Exterior moulds were created to minimise geometry distortion during separation (*Goodship and Birch, 2005*).

### Measurement of sub-tendon mechanical properties

The CSA of each sub-tendon was measured (ImageJ, v1.51) at the same level as the thinnest region (~10 to 40 mm proximal to insertion) of the whole Achilles tendon. Each sub-tendon was tested using a mechanical testing system (5967, Instron, Norwood, MA) with a linear electric motor and a 3 kN load cell. After secured into cryoclamps with 60 mm gauge length, a pre-load was applied (10 N for gastrocnemii and 20 N for soleus) and then 20 cycles of pre-conditioning (triangular wave to 5% strain) was applied. The load was removed to allow the specimen to go slack (4 mm) before pulling to failure (rate: 0.75 mm/s). Young's modulus and stiffness were calculated as the slope of the linear region of the stress–strain and force–displacement curves.

### Construction of Achilles sub-tendon computer models

After separation, the three sub-tendons were carefully assembled and fitted into the original exterior mould. Cross-sections (10 mm apart) were cut through the mould and the sub-tendons from proximal to distal, starting from the soleus musculotendinous junction. These cross-sections were then photographed together with a ruler (1 mm accuracy) using a 12-megapixel camera yielding a resolution of 0.014 mm/pixel. Since the fascicles intertwined at the insertion, we could only confidently separate sub-tendons to a level approximately 10 mm proximal to the calcaneus, thus the reconstructed models end at this point.

Three separate Achilles sub-tendon models were created in a web-based computer-aided design platform (Onshape, PTC, Boston, MA). First, each sub-tendon cross-section was delineated (perimeter, then internal borders) and aligned according to the longitudinal reference line. The CSA differences between the drawn sub-tendon area and the photographic measurements were less than 1 mm$^2$. Next, the cross-sections belonging to each sub-tendon were constructed from distal to proximal, creating three solid parts. The models constructed were inspected to ensure no penetration or separation between each part before conducting simulations.

### Finite element analysis

A commercially available FEA software (ANSYS v19.0, Ansys Inc, Canonsburg, PA) was used for mesh generation and static stress analysis. Each sub-tendon was modelled as a hyperelastic, non-linear, neo-Hookean material (*Handsfield et al., 2017*; *Hansen et al., 2017*) with the initial modulus (Lat. G: 226.7 MPa, Med. G: 143.2 MPa, Soleus: 103.1 MPa) obtained by curve-fitting the in vitro axial tensile testing results. Sub-tendons were meshed into quadratic 10-node tetrahedral solid elements (TET10). A mesh convergence test was conducted using static stress analysis with criteria of 1% for each model. Maximal element edge length for each model was 0.8, 0.5, and 0.4 mm,

respectively, and yielded 258402, 195286, and 223423 elements, respectively. The contact faces between sub-tendon pairs were manually selected and assigned different contact properties: frictionless, frictional (with coefficient 0.2, 0.4, 0.6, 0.8, and 1.0) and bonded. Surface-to-surface translational joints, with no restriction along the longitudinal axis while limiting transverse plane separation, were assigned at the contact faces to prevent sub-tendon separation. Fixed displacement at distal faces (to mimic the fixed calcaneus insertion) and individual 100 N linear ramped tensile forces at the proximal face of each sub-tendon (to mimic muscular pull) were applied.

### In vivo muscular stimulation study – participants

This study was approved by the University College London Research Ethics Committee (ref. 1487/1001). All participants were well informed and provided consent in accordance with the Declaration of Helsinki. Participants were assigned into two groups according to their age (Young: 20–30 years, n = 9; Old: over 50 years, n = 7). The exclusion criteria were having any previous diagnosed Achilles tendon injuries and any systematic disease that may affect muscle-tendon functions.

### Experiment procedures

The participants were prone on an examination bed with feet firmly step against the wall (~90°). Three pairs of 50 mm diameter electrodes were placed on the motor points of the three muscles bellies (*Botter et al., 2011*; *Kim et al., 2005*). The skin was cleaned with alcohol swaps and shaved if necessary to ensure minimal intensity for inducing muscular contraction and minimising discomfort. Each muscle was stimulated using a commercially available transcutaneous electrical neuromuscular stimulation device with 30 Hz, 300 µs pulse train, and an on–off cycle of 8 s (4 s on and 4 s off) for 2 min to induce tetanic contractions. Before the stimulation, an ultrasound probe (2–8 MHz, 9L-D, LOGIQ S8, GE Healthcare, Buckinghamshire, UK) was placed at the musculotendinous junction of the three muscles to ensure proper image quality. During the stimulation, the intensity increased gradually (to ~20 mA) until visible displacement of the stimulated muscle (e.g. medial gastrocnemius) was recorded, while at the same time, no visible contraction of other muscles (e.g. lateral gastrocnemius and soleus) was observed, before moving the probe to the soleus musculotendinous junction to measure its displacement (*Figure 7*). At least three consecutive contractions were recorded and stored for offline analyses. Randomised sequence and ample rest between stimulations were used to reduce artefacts.

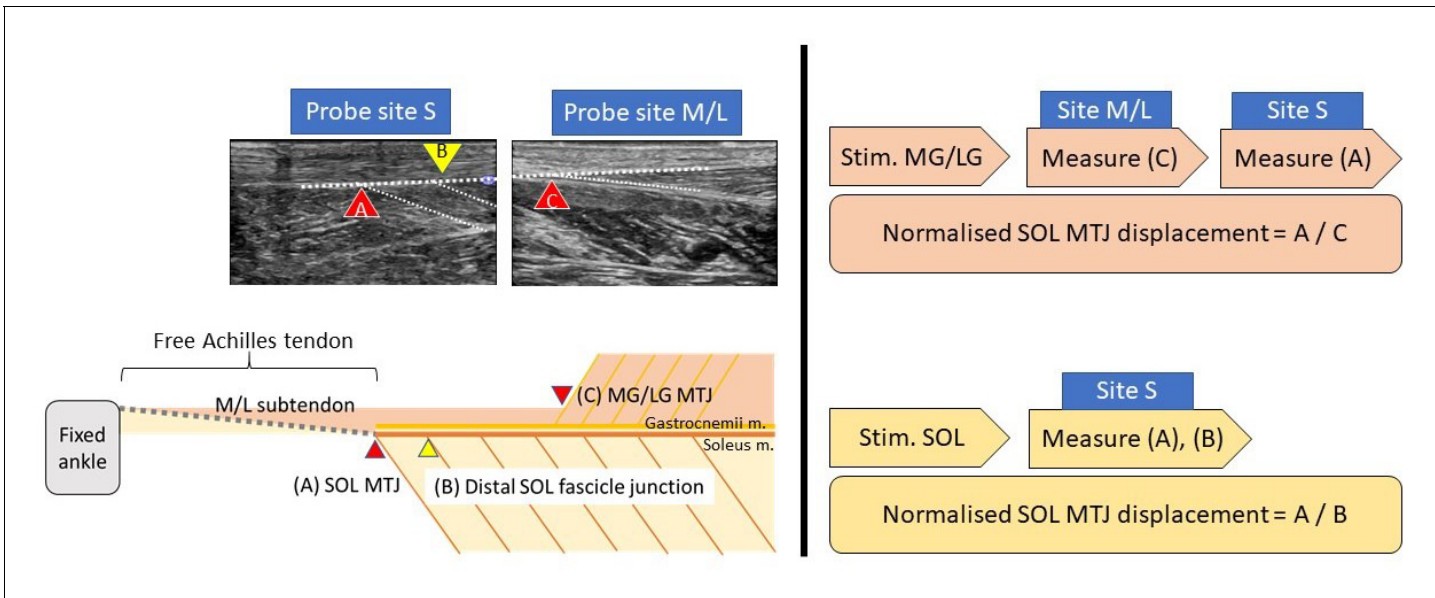

**Figure 7.** Simplified sagittal plane view of triceps surae muscle-tendon units and corresponding ultrasound measurement locations (left) and sequences of measurement during each stimulation trial (right). MG/LG: medial or lateral gastrocnemius, SOL: soleus. MTJ: musculotendinous junction.

## Ultrasound image analysis

The analysis was performed in a semi-automatic way using an in-house MATLAB code (R2019a, MathWorks, Natick, MA), utilising the Computer Vision Toolbox add-on for tracking displacements and correcting movement artefacts. The peak soleus musculotendinous junction displacement of each stimulation trial was tracked and normalised by its contracted muscular displacements (for gastrocnemii: musculotendinous junction, for soleus: one distal muscle fascicle–aponeurosis junction; *Figure 7*, left), assuming the muscular displacement affects only by stimulation intensity, while subtendon displacement affects by interface sliding capacity. We arbitrarily chose the distal soleus muscle fascicle–aponeurosis junction as the reference point (*Muramatsu et al., 2001*) since no clear anatomical landmark lies within the soleus muscle.

## Statistical analysis

Statistical analysis was performed using SPSS (v26, IBM, Armonk, NY). Kruskal–Wallis H tests were used for comparing mechanical properties among sub-tendons, and Mann–Whitney U tests were used for post hoc analysis. In vivo soleus junction displacement between groups was compared using Mann–Whitney U tests. Level of significance was set at 0.05, two tailed. A minimum of total 12 participants for the in vivo study was determined by a priori sample size estimation (d = 1.9, $\alpha$ = 0.05, $\beta$ = 0.90) using G*Power 3.1.9 (Universität Düsseldorf).

## Acknowledgements

We thank Dr Diana Corben for technical assistance with mechanical testing. We thank Prof Dylan Morrissey, Prof Hazel Screen, and Mr Gamalendira Shivapatham for their assistance in developing ultrasound methodology. NY thanks the generosity of the Taiwanese government in providing a scholarship for his PhD studies.

## Additional information

### Funding

No external funding was received for this work.

### Author contributions

Nai-Hao Yin, Conceptualization, Formal analysis, Methodology, Writing - original draft; Paul Fromme, Validation, Methodology, Writing - review and editing; Ian McCarthy, Conceptualization, Validation, Methodology, Writing - review and editing; Helen L Birch, Conceptualization, Resources, Validation, Methodology, Writing - review and editing

### Author ORCIDs

Nai-Hao Yin  https://orcid.org/0000-0002-6764-9790
Paul Fromme  http://orcid.org/0000-0001-5992-2526
Helen L Birch  https://orcid.org/0000-0002-7966-9967

### Ethics

Human subjects: The study was approved by University College London Research Ethics Committee (ref. 1487/1001). All participants were well informed and provided consent in accordance with the Declaration of Helsinki.

### Decision letter and Author response

Decision letter https://doi.org/10.7554/eLife.63204.sa1
Author response https://doi.org/10.7554/eLife.63204.sa2

## Additional files

### Supplementary files
• Supplementary file 1. Statistical analysis outcome of three sub-tendon mechanical properties. (File separated uploaded).

• Transparent reporting form

### Data availability
Data analysed during this study are included in the manuscript.

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
