## [Decision Letter]

**Acceptance summary:**

This manuscript elegantly highlights the oversimplification of current Achilles tendon functional analyses. The current findings provide data on how the anatomy and changes with age predispose to tendon injuries.

**Decision letter after peer review:**

Thank you for submitting your article "Individual variation in Achilles tendon morphology and geometry changes susceptibility to injury" for consideration by *eLife*. Your article has been reviewed by three peer reviewers, including Carlos Isales as the Reviewing Editor and Reviewer #1, and the evaluation has been overseen by Mone Zaidi as the Senior Editor. The following individual involved in review of your submission has agreed to reveal their identity: Josh Baxter (Reviewer #3).

The reviewers have discussed the reviews with one another and the Reviewing Editor has drafted this decision to help you prepare a revised submission.

Summary:

In this manuscript examines the impact of Achiles tendon composition and geometry on biomechanics characteristics. Biomechanical and computer modeling approaches have been utilized to conclude that there is significant individual variation that could be predictive of risk for tendon injury. Results highlight the importance of considering the behavior of the sub-tendons, which are connected to the two heads of the gastroc muscle and the soleus muscle, in properly understanding the overall behavior of the Achilles tendon. Relevant to the field as a whole, a key finding from the present study is demonstrating that the widespread approach of using the soleus musculotendinous junction to measure in vivo tendon mechanical properties is not necessarily appropriate, because with aging the influence of the gastrocnemius sub-tendons becomes more prevalent.

Essential revisions:

1) The authors state in the Results that tendons from 3 males and 2 females were used for study, but Table 1 indicates 2 males and 3 females. This needs to be reconciled.

2) In the Results and Discussion, the authors continually emphasize the differences in mechanical properties between the three sub-tendons in their in vitro mechanical loading studies (Table 1). However, it is important to note that the only real difference between the three sub-tendons is their cross-sectional area, and indeed, this is likely the only contributor to their difference in mechanical properties. Notably, the structural mechanical properties of failure force and stiffness do show differences between the gastroc and soleus-related sub-tendons, but their material property equivalents (which are corrected for differences in geometry) of ultimate stress and Young's modulus do not. This result reinforces that the only major difference here is in geometry, and as written, this important fact is obscured by the authors' interpretations of their own data.

3) Given the importance of size in the in vitro mechanical properties (as mentioned in the comment above), I am surprised that the authors made no attempt to account for differences between males and females in their studies, as presumably males would be more likely to have larger CSA for their Achilles tendon (and consequently altered mechanical properties). The possible differences between sexes is not discussed anywhere in the manuscript to my knowledge, but should be.

4) The choice of samples for the modeling studies is quite skewed, and this limitation is not acknowledged anywhere. The authors have used two male tendons from donors in their mid 50s, and one tendon from a teenaged female. The inclusion of the young female tendon for modeling is a bizarre choice and nowhere explained nor rationalized. The authors did include a "young" group for their in vivo studies, but these individuals were aged 20-29 years; therefore the relevance of this one young female tendon used for modeling has questionable relevance to the study as a whole. The study would be greatly enhanced by the exclusion of this abnormal sample, and replacing it with one better matched to their in vivo cohort of data (since the authors attempt in many locations to establish links between behaviors seen via FEA modeling and behaviors seen during in vitro and in vivo testing).

5) The mechanical testing results are surprising. Specifically, the large variations between sub-tendons material properties (Young's Modulus) shows up to a 5-fold difference in elastic modulus between the lateral and medial gastrocnemius for specimen 78-F. Is this a real tendon property or a measurement artifact caused by the decision to calculate tendon stiffness by dividing maximal force by maximal displacement. Why not quantify the linear part of the force-displacement curve? This methodologic decision could have major implications on the results and ultimately the conclusions. Please address.

6) I do not think linear regression is an appropriate statistical test to determine the effect of age on tendon displacement when the ages are clustered around 2 age groups – ~ 25 and ~ 60 years old. Consider other statistical tests that are more appropriate.

7) Some of the differences between tendons in the computer simulations appear to be caused by differences in geometry (given the material properties are the same). However, with the small sample size (n=3) and the huge variations in tendon displacements between models, I question the utility of stating that geometry impacts tendon mechanics. Generally, in musculoskeletal biomechanics, we accept that form governs function. Providing more insight into the effects of geometry on displacements (directionality, magnitude, etc) would be more impactful.

8) The in vivo experiment found decreased tendon displacement in the older adults during involuntary contractions. However, it did not appear that these contractions were normalized for the magnitude of load carried by the tissue. It seems plausible that the older group generated less tendon load, which could explain decreased tendon displacement. Please address.

9) While this study is an important next step towards understanding the complex mechanics of the sub-tendons during locomotion and ultimately pathology progression, it is unclear how translational the current findings are. Are sub-maximal contractions isolated to individual muscles physiologically relevant? What do you think would change if the soleus was partly activated while the gastrocnemius muscles were fully activated? I understand this is technically and logistically difficult for human subjects research. However, this seems like something well positioned for the computational model. Please reframe the physiological impact within the current loading paradigm.

10) Expand on how the in vivo experiments build confidence in the computer model. The incremental decrease in soleus tendon displacement as friction between the sub-tendons increases makes conceptual sense. Did you try to use literature reports of aponeurosis material properties to inform a “physiologic” condition? Where do you think young and old cohorts fall along this sub-tendon frictional continuum?

11) Stiffness being the compensatory mechanism for stronger soleus muscle is not compelling. Each of the 3 sub-tendons had similar material properties so it makes sense that tendon stiffness and cross-sectional area scaled with muscle strength. I think this is where a computer model provides a powerful platform to investigate how the complex interaction between geometric parameters and material property variations impacts tendon function.

---

## [Author Response]

Essential revisions:1) The authors state in the Results that tendons from 3 males and 2 females were used for study, but Table 1 indicates 2 males and 3 females. This needs to be reconciled.

Thank you for spotting this error. We have corrected the text to “2 males and 3 females”.

2) In the Results and Discussion, the authors continually emphasize the differences in mechanical properties between the three sub-tendons in their in vitro mechanical loading studies (Table 1). However, it is important to note that the only real difference between the three sub-tendons is their cross-sectional area, and indeed, this is likely the only contributor to their difference in mechanical properties. Notably, the structural mechanical properties of failure force and stiffness do show differences between the gastroc and soleus-related sub-tendons, but their material property equivalents (which are corrected for differences in geometry) of ultimate stress and Young's modulus do not. This result reinforces that the only major difference here is in geometry, and as written, this important fact is obscured by the authors' interpretations of their own data.

We agree that our data show no difference in material properties between the sub-tendons. Where we refer to differences in mechanical properties, we mean the mechanical properties of the sub-tendon as a whole rather than the properties of the material. However, in the light of the reviewer’s comment and on re-reading our manuscript we agree that this is not clear and may be confusing to the reader. When we refer to “geometry” we mean to include in that the CSA of the sub-tendons and the orientation (amount of twist) and again we can see that we have not made this clear enough in the text. We have modified the manuscript (see Introduction, Results, Discussion, Materials and methods) to explain what we mean by geometry and to better emphasise the importance of tendon geometry in influencing the mechanical behaviour.

3) Given the importance of size in the in vitro mechanical properties (as mentioned in the comment above), I am surprised that the authors made no attempt to account for differences between males and females in their studies, as presumably males would be more likely to have larger CSA for their Achilles tendon (and consequently altered mechanical properties). The possible differences between sexes is not discussed anywhere in the manuscript to my knowledge, but should be.

This is a valuable comment and an interesting point, especially as epidemiology studies suggest a difference in the incidence of Achilles tendon injury between the male and female genders across the life course. The aim of our study was to investigate the impact that morphological diversity has on sub-tendon behaviour rather than attribute this to male or female gender. Leading on from our work, future studies should be conducted to specifically test for differences between genders and age-related changes to sub-tendon behaviour. We have added text to the Discussion to cover this point.

4) The choice of samples for the modeling studies is quite skewed, and this limitation is not acknowledged anywhere. The authors have used two male tendons from donors in their mid 50s, and one tendon from a teenaged female. The inclusion of the young female tendon for modeling is a bizarre choice and nowhere explained nor rationalized. The authors did include a "young" group for their in vivo studies, but these individuals were aged 20-29 years; therefore the relevance of this one young female tendon used for modeling has questionable relevance to the study as a whole. The study would be greatly enhanced by the exclusion of this abnormal sample, and replacing it with one better matched to their in vivo cohort of data (since the authors attempt in many locations to establish links between behaviors seen via FEA modeling and behaviors seen during in vitro and in vivo testing).

We are constrained in our studies by the availability of fresh whole human Achilles tendons suitable for dissection to describe morphology. However in this study we were aiming to cover a wide range of the different morphologies that exist between individuals so that we could assess how these differences impact on sub-tendon movement relative to each other with changing friction between the sub-tendons. We were not aiming to produce one model that would describe a particular gender/age/type of individual. We therefore do not see this as a limitation of our work but more of a limitation is that there may be other morphologies that we haven’t covered. We have added sentences to the Results section) to acknowledge that we selected these specimens due to availability and they should only be used as an indication of how tendon shape and rotation can vary between each individual, regardless of age and gender.

5) The mechanical testing results are surprising. Specifically, the large variations between sub-tendons material properties (Young's Modulus) shows up to a 5-fold difference in elastic modulus between the lateral and medial gastrocnemius for specimen 78-F. Is this a real tendon property or a measurement artifact caused by the decision to calculate tendon stiffness by dividing maximal force by maximal displacement. Why not quantify the linear part of the force-displacement curve? This methodologic decision could have major implications on the results and ultimately the conclusions. Please address.

We agree that our data show a surprising variation in elastic modulus between the specimens. We also agree that giving the modulus in the linear part of the force-displacement curve would be better and we have recalculated the modulus and stiffness as the slope of the linear region in the stress-strain and force-displacement curve and modified both Results and Materials and methods sections (Table 1). The modelling results are not affected by this since we used curve-fitted raw data to estimate the hyperelastic properties of sub-tendons. Even with this re-calculation the Young’s modulus shows a 4.4-fold difference between the lateral and medial gastrocnemius for specimen 78-F, which we are not able to explain. We believe this is a real difference and not an artefact and may relate to unidentified tendon pathology.

We were unfortunately not able to obtain medical records of the donors so further investigation is not possible.

6) I do not think linear regression is an appropriate statistical test to determine the effect of age on tendon displacement when the ages are clustered around 2 age groups – ~ 25 and ~ 60 years old. Consider other statistical tests that are more appropriate.

We used the Mann-Whitney U test (see Materials and methods) to compare the two group means (Discussion). The scatter plot and description relating to regression has been removed.

7) Some of the differences between tendons in the computer simulations appear to be caused by differences in geometry (given the material properties are the same). However, with the small sample size (n=3) and the huge variations in tendon displacements between models, I question the utility of stating that geometry impacts tendon mechanics. Generally, in musculoskeletal biomechanics, we accept that form governs function. Providing more insight into the effects of geometry on displacements (directionality, magnitude, etc) would be more impactful.

We used the term “geometry” to indicate the CSA, shape, rotation and arrangement of the three sub-tendons relative to each other. For clarification, we have modified the term in the manuscript (see Introduction, Results, Discussion, Materials and methods) to better represent the complex sub-tendon structure. We think that this modification will emphasise better our point that geometrical features (twisted structure with variable CSAs along the length) are more influential to tendon displacement than material properties and mechanical properties of the isolated structures.

We have also included a new figure (Figure 2—figure supplement 1) and a sentence in the Results to demonstrate the differences in transverse plane movement of each model, showing different tendon geometries produce distinct rotational patterns.

8) The in vivo experiment found decreased tendon displacement in the older adults during involuntary contractions. However, it did not appear that these contractions were normalized for the magnitude of load carried by the tissue. It seems plausible that the older group generated less tendon load, which could explain decreased tendon displacement. Please address.

We were not able to measure the muscle force generated by the involuntary contractions and it is indeed plausible that the older group generated less muscle force and therefore tendon load than the younger group. We have therefore expressed our data as values normalised to muscle displacement to overcome this possible difference. As stated in the Materials and methods section (Figure 7) in order to normalise the individual differences, we selected the gastrocnemii muscle-tendon junction (C) or soleus muscle fascicle-aponeurosis (B) to measure muscular displacements caused by stimulation, with the hypothesis that proximal displacement is highly related to the active muscular contraction while having little influence from distal passive sub-tendon properties. Therefore, stiffening of the interface between gastroc tendons and soleus MTJ (A) would reduce A relative to B when the soleus is stimulated resulting in a decreased ratio (as seen in our results) but would increase A relative to C when the lateral or medial gastroc is stimulated and increase the A/C ratio (as our results suggest but not significant).

9) While this study is an important next step towards understanding the complex mechanics of the sub-tendons during locomotion and ultimately pathology progression, it is unclear how translational the current findings are. Are sub-maximal contractions isolated to individual muscles physiologically relevant? What do you think would change if the soleus was partly activated while the gastrocnemius muscles were fully activated? I understand this is technically and logistically difficult for human subjects research. However, this seems like something well positioned for the computational model. Please reframe the physiological impact within the current loading paradigm.

These are interesting questions and our computational model is suitable to address these in part, in combination with human motion data. This represents a significant body of work and is the direction we are taking with our research efforts. We have modified our conclusion and cut down on suggesting how these results may translate into the clinic as we appreciate that this is somewhat premature at this stage of our work.

10) Expand on how the in vivo experiments build confidence in the computer model. The incremental decrease in soleus tendon displacement as friction between the sub-tendons increases makes conceptual sense. Did you try to use literature reports of aponeurosis material properties to inform a “physiologic” condition? Where do you think young and old cohorts fall along this sub-tendon frictional continuum?

Our in vivo data showed a consistent and significant decrease in soleus tendon displacement in the older age group when soleus muscle was stimulated. In our modelling experiments all three models predicted this result. In contrast, the different models showed different responses when considering the impact of changing sub-tendon frictional contact on soleus tendon displacement when the medial and lateral gastrocnemius muscles were stimulated. Our in vivo data also displayed this variability with greater standard deviations and no significant differences between the young and old age groups. This agreement between our models and in vivo measurements provides confidence in our modelling experiments.

We focused on the “free Achilles tendon” region below the aponeurosis of the gastrocnemius and attachment to the soleus muscle to avoid the complex anatomy of the muscle tendon unit in this more proximal region, as we are unable to measure these “proximal” influences. This limitation is acknowledged in the Discussion.

Our in vivo and FEA results provide good evidence that interface stiffness (by induced friction) increases with age and suggest the best fit is between 0 and 0.8.

11) Stiffness being the compensatory mechanism for stronger soleus muscle is not compelling. Each of the 3 sub-tendons had similar material properties so it makes sense that tendon stiffness and cross-sectional area scaled with muscle strength. I think this is where a computer model provides a powerful platform to investigate how the complex interaction between geometric parameters and material property variations impacts tendon function.

We agree that a greater CSA rather than a stiffer material achieves the higher stiffness of the soleus sub-tendon. We have removed some text and modified to make this clear (Discussion).